# Putting Research to Action: Integrating Collaborative Governance and Community-Engaged Research for Community Solar

**Emily Prehoda \*, Richelle Winkler and Chelsea Schelly** 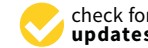

Department of Social Sciences, Michigan Technological University, 1700 Townsend Drive, Houghton, MI 49931, USA; rwinkler@mtu.edu (R.W.); cschelly@mtu.edu (C.S.)

\* Correspondence: ewprehod@mtu.edu

**Abstract:** Community solar involves the installation of a solar electricity system that is built in one central location with the costs and benefits distributed across voluntary investors who choose to subscribe and receive credits based on the generated energy. Community solar is gaining attention because of its potential to increase access to renewable energy and to democratize energy governance. This paper reflects on community-engaged research experiences in two rural community case studies in Michigan, USA, focusing on obstacles that were experienced during the research process rather than empirical findings from the research. We highlight difficulties we experienced to help advance a conceptual argument about incorporating collaborative governance strategies to improve community-engaged research for community energy projects. Our reflections illustrate challenges in community-engaged research that are associated with identifying who should be included in the decision-making process, sustaining participation and avoiding exploitation, establishing and communicating final decision-making power, and giving attention to outputs and outcomes of the research. We argue that collaborative governance strategies can help to address these challenges, as we experienced firsthand in our project.

**Keywords:** community solar; community engaged research; collaborative governance; disadvantaged

## 1. Introduction

The U.S. energy system is currently undergoing a transition to include increasing amounts of renewable energy that distributed generation powers. Energy transitions are characterized by a significant set of long-term structural changes to the patterns of energy use in society, which can have a significant impact on quality of life, economic organization, and the activities and practices of individuals (Sovacool et al. 2016). Community members have an important stake in how energy transitions occur; however, they do not often have much say in when, where, or how renewable energy projects are built (Catney et al. 2014). Engaging communities in these processes has several potential benefits (Kim 2017). It can reflect local interests and priorities (Petersen 2016), keep economic gains from energy savings local (Magnani and Osti 2016), build community pride and cohesion (Burchell et al. 2016), and help to create awareness and transparency on energy issues that may be unclear or confusing (Rogers et al. 2012).

One increasingly popular way that communities can be directly involved in energy transitions is through community solar. Community solar involves a solar electricity system being built in one central location, while the costs and benefits are distributed across voluntary investors who choose to subscribe and receive credits based on the generated energy. Community solar is gaining attention because it aims to democratize energy by bringing ownership and control of energy generation to a large

number of people (NREL 2018; Hoffman and High-Pippert 2015). Local community members can then become personally invested through a common interest in local energy generation. Community solar provides a forum for awareness, education, and discussion regarding how energy systems can work on a local scale (Klocke et al. 2017). Finally, community solar attempts but ultimately struggles to promote social responsibility through access and affordability to energy systems (Brummer 2018). For these reasons, agencies and organizations, such as the Department of Energy, the National Renewable Energy Lab, the Solar Energy Industries Association, and the Smart Electric Power Alliance (to name a few), increasingly promote community solar. The number of community solar projects in the U.S. has grown from only 36 kW in 2006 to 1226 MW through Q2 2018 (SEIA 2018).

Still, community solar projects tend to be accessible only to relatively wealthy people (NREL 2018; SEPA 2015; LOTUS 2015), and they are often designed (and ultimately controlled) by the same energy-providing utilities that control our other energy systems (Lerch 2017; Catney et al. 2014). Given these problematics, our project team implemented a community-engaged research project to explore the potential costs, benefits, and local contexts of starting community solar. The research project aimed to give local communities control over the process of deciding whether or not to build a community solar system, and, if so, how to design a program that would elicit broad interest and be affordable and accessible to low-to-moderate income households.

The purpose of this paper is twofold. First, we critically reflect on the community-engaged research process that we employed in two case communities in the Upper Peninsula of Michigan to illustrate how community-engaged research (CER) can insert more local control and affordability into community energy systems. Second, we use these reflections to advance a conceptual argument about some of the challenges that community-engaged research faces and how incorporating principles of collaborative governance may help to address those limitations. We contend that CER can incorporate principles of collaborative governance to become better equipped for community solar program development.

## 2. Background and Literature Review

### 2.1. Community Energy

Community energy projects are increasingly being promoted as a path toward renewable and decentralized energy structures that will help to promote a more sustainable and resilient society while offering communities legitimacy, consensus, and voice (Barr and Devine-Wright 2012). Community energy projects aim to pay specific attention to 'community' or, in other words, who develops and controls the project, who is impacted by the project, and how they are impacted (Walker and Devine-Wright 2008). The community energy literature stresses keeping local control and operation when developing community energy projects as well as keeping benefits local (Catney et al. 2014). Some community energy projects are conceptualized as grassroots initiatives that utilize local leaders and stakeholders to represent the local situation, interests, and values of the involved community (Seyfang and Smith 2007). In reality, however, community energy projects are often subject to external motivations, management, and control, meaning that they are not always community engaged or driven (Catney et al. 2014).

Martiskainen et al. (2018) argue that community-engaged energy projects are inherently political. National community energy initiatives use the tactic of emphasizing the benefits of local energy generation; however, control of these energy systems still remains in federal governance structures or powerful decision-makers, rather than the communities within which they operate (Smith 2005; Walker et al. 2007). Community energy projects tend to lack a unifying vision as there can be tensions between who spearheads the project versus who participates in designing and implementing the project (Catney et al. 2014). Some community energy projects rely on centralized government funding from initiatives that articulate local energy in national energy policy (Walker and Devine-Wright 2008; Catney et al. 2014; Walker et al. 2010). People are viewed as objects rather than subjects of change in

the energy infrastructures of these communities. These community energy projects continue to support individual rather than collective strategies for project success (Cameron 2010; Catney et al. 2014). This means that there is an emphasis on individual behavior change towards some predetermined goal, which can be defined without or with minimal community input (Martiskainen et al. 2018; Batel et al. 2013; Maniates 2001). Many local communities have difficulty accessing decisions regarding energy systems that can adversely impact their communities (Martiskainen et al. 2018; Rau et al. 2012). For example, in our project, participants repeatedly described a wind project that was initiated by a larger energy development firm that ignored community input and values.

Substantial resources (e.g., human, financial, political, social, and built capital) are required for communities to really control the process of deciding on, developing, and implementing a community energy project. Rural (or otherwise structurally disadvantaged) communities may either lack these resources or not be in a position to devote limited resources to investigating community energy potential. There may be internal barriers, such as a lack of knowledge regarding energy systems, skills to navigate governance and political structures, or monetary resources that can affect community energy project success (McKenzie-Mohr 2000; Dóci and Vasileiadou 2015). Moreover, communities may struggle to engage and maintain the role of civil society in the decision-making process (Batel et al. 2013).

Our research focused specifically on community solar as a community energy project. However, community solar is not limited to one specific model, as three community solar types dominate the growing field: utility-scale, non-profit, and special purpose entity. In the utility-sponsored model, utilities build, own, and operate the system. Ratepayers can voluntarily participate by contributing a payment (upfront or ongoing) to support the system (NREL 2010). In the second model, non-profit organizations partner with the surrounding community or businesses who can provide donations to finance the project. Donors in this type do not receive direct benefits from the system, but do share in indirect benefits through tax deductions and social benefits (NREL 2010). In the final model, individuals, groups, or organizations come together to form a small business to take advantage of commercial tax benefits that accompany solar photovoltaic (PV). Benefits from this model can be realized by the organizers themselves or in a partnership between a community and special-purpose entity (NREL 2010).

How to develop and design a community solar project ultimately depends on the enabling policy context, which varies by state in the U.S. For example, some states (i.e., California, Minnesota, and Maryland) have formal laws that allow community solar program implementation directed by various different actors. Michigan does not, which leaves community solar program development to the utility's discretion. In Michigan, most community solar programs are spearheaded by members of an electric cooperative (e.g., Cherryland Electric) or by municipal utilities (e.g., the Traverse City Board of Power and Light, and the Marquette Board of Light and Power) (GLREA 2013). The Consumers Energy Company, an investor-owned utility, owns and operates a community solar program in the lower peninsula. The key takeaway here is that, in Michigan, a community solar program relies on and requires the ability to partner with a utility to install panels, establish leases, sell PV power, and/or ensure sound investments. More innovative solutions may be necessary to encourage change in community solar policies, laws, and adoption (Klein and Coffey 2016); however, this project focuses on community solar development within the context of the existing electrical energy policy regime.

Regardless of the policy context, we believe that community-engaged research (CER) could help to improve the process of community energy project development. Collaborative partnerships between communities, research institutions, and utilities designed around the principles of CER can arguably bring the necessary resources while preserving community control and decision-making in the community energy process. The process could empower community members to speak out about potential impacts of a local energy project and begin to take ownership by participating in the program's design. Ultimately, CER projects could help decision-makers to develop community energy projects that reflect community beliefs, goals, and values.

### 2.2. Community-Engaged Research

Community-engaged research improves the meaningful participation of community members by creating collaborative spaces between community members, community organizations, and academic researchers to address community issues or problems (Bhattacharjee 2005; Learned et al. 2017; Duran et al. 2013; Kantamneni et al. 2019; Klocke et al. 2017). We use the term "community-engaged research" or (CER) as synonymous with "community-based participatory research (CBPR)", "participatory action research (PAR)", and other similar concepts. Following CER principles, community members play an important role in determining the trajectory of the research questions, project design, and data collection and analysis. Action research in particular emphasizes the goal of improving community practices and empowering community members in addition to increasing knowledge (Stoecker 2012; Bradbury-Huang 2010; Ferrance 2000).

CER processes can improve the relevance of research, ensuring that projects are important to communities and benefit communities (Israel et al. 2017; Hacker 2013; Strand et al. 2003; Wallerstein and Duran 2010). Still, CER is a relatively new practice in most fields (outside of public health) and particularly for community energy projects. Our team has not been able to find any other published work that explicitly uses community-engaged research principles for community energy projects. As a relatively new (and different!) endeavor, several articles, chapters, and briefs offer protocols and principles for conducting CER (e.g., Burns et al. 2011; Israel et al. 2017; Strand et al. 2003). Yet, in practice, CER still faces some important challenges for teams to grapple with that are not easily answered with a clear principle or protocol.

One common CER challenge is defining the scope of the community with whom researchers collaborate (Long et al. 2016; Kantamneni et al. 2019). Contentions lie in whether to define communities based upon geography, different demographics, common interests, or community identity (Agrawal and Gibson 2001; Long et al. 2016). Who are the partners? Who is represented and how? What are their various roles? (Goold et al. 2016). Ultimately, who is included, and to what degree, drives the direction of the research, representation in the data, and likely outcomes (Hibbard and Madsen 2003, 2004). While CER scholarship recognizes this complication, it has not been fully resolved. CER principles suggest that all stakeholders should be included as partners at the table; however, including every affected individual is not feasible. Projects that seek community representatives (Goold et al. 2016) to serve as the voice of a broader community (Stoecker 2012) may function to empower those who are already relatively powerful, leaving out the most disenfranchised voices (Tumiel-Berhalter et al. 2005). CER principles note the importance of forming a collaborative, equitable partnership, but they fall short on providing clear indications of who should be involved in the partnership, under what conditions, and how.

A second challenge to community-engaged research is sustaining participation as a result of a history of exploitation within the community (Morris 2017). This is particularly difficult to overcome in disadvantaged and low-to-moderate income communities (El Ansari and Weiss 2005). Many disadvantaged communities, and particularly tribal communities, have experienced a history of research abuse and projects that have done little to benefit their communities (Israel et al. 2017; Hacker 2013). CER is specifically designed to combat community exploitation by offering community members the opportunity to participate and collaborate in research that will empower participants and be directly used for the community's benefit. The Department of Energy's SunShot Initiative and Solar in Your Community Challenge attempts to expand access to, and the affordability of, solar PV in these communities. Still, the time and effort required of community members to participate as full collaborators in research projects is immense (Baker et al. 1999; Koné et al. 2000). This can be especially troublesome in disadvantaged and low-to-moderate income communities (El Ansari and Weiss 2005; Flicker et al. 2007; Adhikari et al. 2014; Tosun 2000). Requesting this effort may inadvertently result in another form of community exploitation: taking people's time without being able to guarantee results. This creates an ethical dilemma (Long et al. 2016), and also can create challenges in recruiting

and maintaining sustained participation among community members who often have many other competing time demands.

CER, and especially Action research, comes from the perspective of undertaking research with the purpose of facilitating social change (Carr and Kemmis 2003). Yet, CER scholarship focuses almost entirely on the process of conducting research, with little attention to how teams use the research results to make decisions or facilitate change. Action research aims to disrupt existing power relations by specifically shifting the role of research participants to active contributors helping to shape knowledge about their community and its problems, and then using this knowledge to push for change (Cawston et al. 2007; Kimura and Kinchy 2016). Scholars provide valuable roadmaps for partnering with community members to collect and analyze data, interpret results, and report out (Hacker 2013; Balazs and Morello-Frosch 2013; Israel et al. 2017); yet it provides little direction regarding what to do with this information to ultimately improve community conditions. The process of engaging in research can be empowering, but even engaged research is not always easily translated into action and may not lead to changes in programs, services, or access.

### 2.3. Collaborative Governance

Collaborative governance (CG) is a decision-making and management approach whereby multiple stakeholders at various levels or scales "engage in consensus-oriented decision-making" (Ansell and Gash 2008, p. 543). CG is generally used to enhance decision-making in policy areas, such as economic development, public health, environmental protection, and land use (Rogers and Weber 2010). It is increasingly being regarded as a strategy to build shared meaning, to learn, and to incorporate change (Innes and Booher 1999). The management of energy systems in the United States is complex and involves actors at the federal (i.e., the Federal Energy Regulatory Commission), regional (i.e., independent system operators), state (i.e., public utility/service commissions), and some local (i.e., municipal utilities) levels. Often, local communities are removed from decision-making regarding energy systems and can be adversely impacted by decisions made at other, higher levels (Lerch 2017). CG approaches in community energy systems could help to shift towards more inclusive decision-making and more successful projects. Two similar approaches that also have merit are participatory design and collective impact, which tend to focus on non-formal actors designing the systems they use (Muller and Kuhn 1993; Schuler and Namioka 1993) and collective decision making for behavior change (Kania and Kramer 2011, 2013), respectively. CG specifically focuses on governance strategies to facilitate collective decision-making in policy arenas, which makes it a more appropriate approach in the context of both CER and community solar program design.

CG scholarship has paid some attention to applications in energy systems and transition decisions. Studies focusing on the U.S. show that, while collaborative planning was a strategy used to improve and advance energy systems, they fell short of participant representation and inclusion due to power imbalances (Purdy 2012). Additionally, U.S. collaborative planning strategies typically slow after the planning stage, with a lack of action following the collaborative stage (Pitt and Congreve 2017). Margerum (2002) argues that CG commits participants to implementation. Yet previous employment of CG strategies in energy systems struggled to successfully implement energy system changes. Despite this, we believe that collaborative governance approaches show good potential for planning and decision-making on community energy projects. It is especially helpful for addressing some of the challenges (summarized above) that are associated with community-engaged research, including defining the scope of community collaborators, sustaining project participation, and decision-making to move research into action.

Chrislip and Larson (1994) argue that the inclusion of all affected and/or interested stakeholders is necessary for successful collaboration. This is important for propelling the collaboration towards a more democratic process. Not including impacted members can impact the legitimacy (Johnston et al. 2010) of the project, ultimately influencing its viability. A best practice strategy emphasizes a deliberative planning process, which involves an extended group discussion to ensure the inclusion of all affected

stakeholders prior to moving out of the planning stage (Hicks et al. 2008; Roussos and Fawcett 2000; Johnston et al. 2010).

In order for CG collaborations to be successful, issues must have salience for participants (Selin and Chevez 1995). Generating and maintaining participation can be difficult due to the time and effort involved (Emerson et al. 2011). CG's solution to this problem is appropriately compensating collaborative participants for their efforts in the decision-making process (Bingham 2009). This mechanism was employed in the city of Seattle to support citizen engagement during a neighborhood planning initiative in the 1990s (Page 2010).

CG scholarship provides a helpful and practical reminder that some collaboration members will be more responsible for final decision-making than others (Ansell and Gash 2008; Stoker 2004). In some cases, multiple levels of decision-making (where stakeholders at different scales collaborate) allow the process to become more adaptable to change (Newig and Fritsch 2009). Still, there is an ultimate decision-maker(s) of the collaborative deliberation process (Newig and Fritsch 2009). A key step here is to lay out process transparency from the beginning (Ansell and Gash 2008), including which and how these decisions will be made, by whom, and with what input from whom else (Ansell and Gash 2008).

CG emphasizes the outputs of successful collaborations. Outputs might be a report detailing analyses and recommendations from the collaboration team to the final decision-makers (Thomas 2008; Page 2010), a guide book for management strategies (Herrick et al. 2009), or a management program (Kallis et al. 2009). Clearly defining outputs at the start of the CG process is important for effective decision-making (Thomson and Perry 2006; Ansell and Gash 2008), yet collaborative governance strategies tend to overemphasize outputs and ignore outcomes (Koontz and Thomas 2006; Thomas 2008).

## 3. Purpose

This paper critically reflects on a case study experience employing CER principles to inform community solar projects in two rural communities. It is a reflective essay meant to illustrate how applying principles from collaborative governance might improve community-engaged research. Ultimately, we argue that community-engaged researchers can integrate principles from collaborative governance to enhance decision-making for action outcomes. Reflecting on our team's experiences, we recognize challenges we experienced and consider how insights from CER and CG can be combined to ultimately improve community energy projects.

## 4. The Case Study

Academic researchers at Michigan Technological University partnered with community leaders from the villages of L'Anse and Baraga (MI, USA), WPPI Energy (a local energy-supply cooperative utility), and planners at the Western Upper Peninsula Planning and Development Region to explore the social feasibility of starting a community solar project in each community. Each of these actors participated as equitable partners in the research endeavor. In both cases, village administrators were interested in the possibility of starting a community solar project but did not want to move forward without engaging directly with the broader community and learning more about whether local people were interested in such a program and how it might be designed so that it would be accessible and attractive to a broad range of community members. The research team generally followed the principles of community-engaged scholarship (the methods are described in more detail below) to evaluate this social feasibility. The research project idea and specific research questions originated from leaders in the community. Decisions about methods and specific details about how, when, and where to engage in research were made collaboratively. Academic researchers and a class of students at Michigan Technological University did the majority of the data collection and analysis. Interpretations were vetted and discussed collaboratively among the full team. Results were shared at public meetings where all local area residents were invited to share their own insights and ideas.

### 4.1. The Communities

The case study sites are the neighboring villages of L'Anse and Baraga, Michigan (Figure 1). We defined our case study community by geographic boundaries. Both villages operate their own municipal electric utilities, both of which are adjacent to the service territory of an investor-owned utility. While some customers in each village receive power from the investor-owned utility, participation in the potential community solar programs can only be offered to the village utility customers. Each village has a population of about 2000 residents, and they are located approximately 3 miles from one another along the Keweenaw Bay on the southern shore of Lake Superior in Baraga County. These are rural and remote communities, located more than three hours away (by car) from the nearest metropolitan area (Green Bay, WI, USA). The cases represent places where community solar projects might be especially challenging. Low-to-moderate income households make up a large proportion of the population in both villages (43% and, 66%, respectively) (MSHDA 2017), which could present a hurdle to participation in community solar given that upfront subscription costs are often substantial and more affluent people are generally more likely to subscribe (LOTUS 2015; SEPA 2015; NREL 2015a, 2015b). These two communities have a large tribal presence, with almost 50% of Baraga's population identifying as American Indian (alone or in conjunction with another race, U.S. Census, ACS 2016). Also, in comparison to very sunny places and to places with high electricity costs, the potential financial return on investment in solar is low here, because there is relatively low solar radiation (3.4–4.4 kWh/m$^2$/day, NREL 2017) in this northern region and because electric rates are near the state and national average ($0.10–0.13/kWh, Village of L'Anse and Village of Baraga Utility[1]). Selling a fairly small amount of solar-produced electricity at moderate rates yields only moderate returns on the investment.

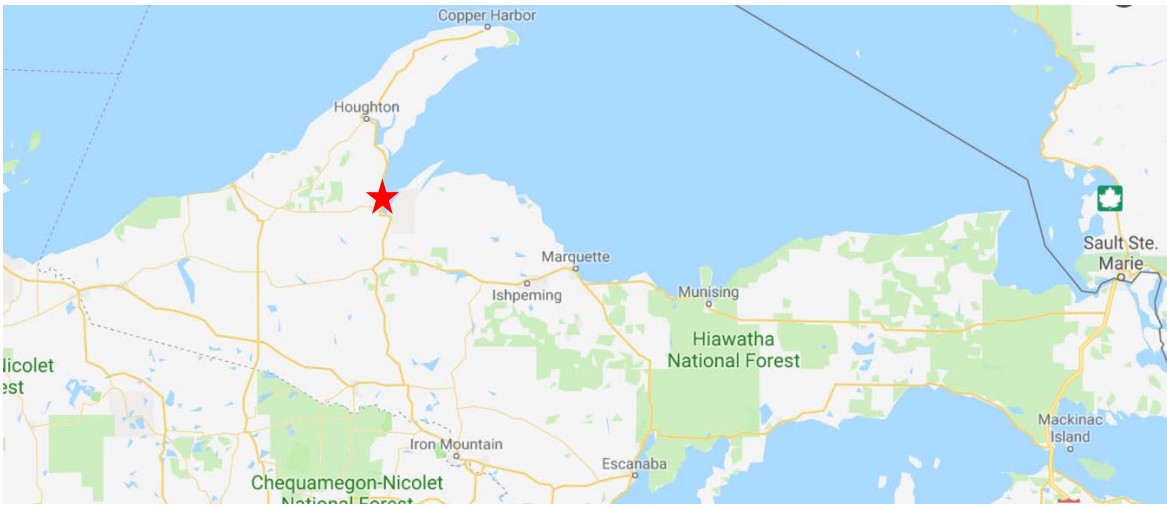

**Figure 1.** The location of the L'Anse and Baraga villages in the Upper Peninsula.

### 4.2. Methods

Data for the social feasibility study included semi-structured interviews with key informants, "world cafe"-style community meetings (Jorgenson and Steier 2013; Brown 2010), a full sample survey in each community, and financial analyses. Each of these steps is described in more detail below. This multifaceted research approach allowed the team to get a sense of the complexities behind support or lack thereof for the proposed community solar program (gleaned from qualitative data) while also making an informed estimate of general program support and projected participation levels (based

---

[1]     These numbers were obtained from personal communication with Village Utility operators.

on quantitative estimates). The semi-structured interviews provided a first glance into potential opportunities and challenges associated with community solar in the local community context, as well as raised awareness about the project idea among community leaders, and helped the team to advertise the first community meeting. Community meetings provided in-depth insights from a broad cross-section of community members, offered people a public chance to express concerns, and offered a unique opportunity to hear how community members talk to one another about the project idea. Because we assume that residents of a small town will discuss the program with one another and that it would be most successful if spread by word-of-mouth, understanding how community members converse with one another was important. The community meeting structure also allowed for data interpretation from community participants responding and reacting to points raised by one another and in the survey results and talking through ideas and themes to ultimately interpret what points are most important locally. The results from the interviews and community meetings both informed the survey design. The primary functions of the surveys were to obtain a basic understanding from a broad representation of community members on support for local community solar, key factors that impact that support, and to estimate the likelihood of participation under different scenarios. Survey results were imperative to demonstrate broad interest to key project leaders. Altogether, this multifaceted approach allowed the team to expand community participation by involving a large number of diverse community members in every research step.

The team started by conducting a critical review of existing community solar programs to draw lessons for successes and failures that we could learn from. We conducted 15 interviews (5 in L'Anse and 10 in Baraga) with community and tribal leaders and social service providers to gain initial insight regarding potential barriers to, or motivations for, local participation in a community solar program. Interviews were audio-recorded for later analysis. We worked with the Keweenaw Bay Indian Community (KBIC) Committee for Alternative and Renewable Energy to determine what initiatives would spark interest in the tribal community and to design data collection strategies that would engage tribal members.

Community meetings offered the general population (beyond the specific research partners) an opportunity for meaningful participation. We held an initial public community meeting in L'Anse to share information about community solar and to facilitate discussion among community members. A total of 49 people attended. The meeting followed a "world cafe" format (Jorgenson and Steier 2013; Schieffer et al. 2004; Brown 2010), where participants were asked to sit at round tables of approximately five people each and to respond in small groups to questions, then to report out to the broader group for general discussion and interpretation. Questions included: (1) what do you like about the idea of L'Anse doing a community solar project? (2) what concerns you most about this idea or makes you think it might not work? (3) would you purchase shares? And why or why not? (4) What are some things that the team needs to consider in designing the program? (5) Do you think that L'Anse should move forward with this? Why or why not? A second community meeting was held three months later where the research team shared survey results (see below), and participants were again asked to discuss community solar possibilities in small groups and report out in a similar format. Notes from each of the small tables as well as the full group discussion were recorded and analyzed for key themes (see Appendix A).

A community survey (designed collaboratively among project partners) was distributed in two separately to all Village of L'Anse and Village of Baraga utility customers to determine community members' interest in participating in a community solar program as well as to provide additional feedback regarding community solar program design. Survey respondents were provided a $5 community currency upon submitting and requesting the $5 as an incentive. Researchers administered community solar surveys through each Village's utility bill. Community members were provided a stamped envelope to return surveys through mail to WUPPDR, or surveys could also be returned to each Village office. Students canvassed door-to-door (convenience sampling) in densely populated neighborhoods to drop off additional surveys face-to-face, answer questions, listen to community

feedback on the project idea, and collect completed surveys as available. We geolocated neighborhoods in the Village of Baraga with low initial response rates and then specifically targeted canvassing efforts there to provide further information about the project as well as give residences additional opportunities to respond to the survey. A total of 174 and 158 Village of L'Anse and Baraga (respectively) utility customers responded to the survey. The response rate was 14% of all residential customers in L'Anse and 24% in Baraga. Survey data were cleaned, coded, and analyzed using the Stata statistical software to estimate support for, and knowledge of, community solar, and to consider variation in these variables by age, income, length of residence, gender, education, and tribal affiliation. This information was then used to help the team (a) estimate who might be likely to participate in the community solar program and (b) understand different demographic needs to improve our community solar program. Additionally, we calculated the predicted number of solar panels that respondents said they would purchase under various program designs in order to understand what kinds of program design were most popular, how they would work for low-to-moderate income residents, and whether a community solar program would likely sell enough shares to be feasible. Survey results were important for demonstrating broad public support for moving forward with a community solar program to research partners, additional village leaders, the electric utility, and the broader community. They also offered a means of participation in decision-making that was (while minimal) accessible to all utility customers (stakeholders).

Finally, the project team integrated results from the interviews, the first community meeting, surveys, and the existing program review to generate program design options that might work for local communities. Research partners reviewed the financial costs associated with building, maintaining, and administering a community solar project of the appropriate size to meet community interest and developed initial models for investment options that both reflected community interests and covered the required costs. The team's goal was to design program options that were both financially sound and affordable and accessible to community members. The team presented initial program options and solicited feedback in the second community meeting, and then adjusted the program design options accordingly. At this writing, L'Anse has decided to start a community project and has started to pre-sell shares following a program design that was generally (with some exceptions) what the project team recommended. The project team in Baraga is still in the process of reviewing and interpreting results.

## 5. Discussion and Reflection

Overall, the project team felt that employing a community-engaged research approach that took participation from a diverse set of community members seriously improved the viability of starting community solar and designing program options that reflect the community's unique interests and needs. A diverse set of local people provided feedback about a community energy project that would be located in their community. This was something that they had not experienced with previous community energy projects. Moreover, the entire project idea was driven by local actors, and the research process helped to build knowledge of community solar, support for the project, and trust in the project development process. Ultimately, the program design incorporated voices from the community that expressed concerns over how people limited by geography or money could participate and what happens to a community solar subscription if you move or no longer want your subscription. The L'Anse program was designed with three different payment/credit options, which together met the interest and needs of the broader community, and it includes ideas from the community, such as collaboration with local non-profit community organizations to facilitate donating solar panels. Overall, the team believes that CER helped to improve the process of designing a community energy program. The community is generally supportive, and the utility is starting to use the process we employed as a model for other communities.

Still, the team encountered challenges that could be addressed by incorporating principles of collaborative governance to inform the community-engaged research process for community solar. We failed at inclusively defining "the community" who participated as core members in our CER process,

and we struggled to sustain broader community engagement. We did not clearly communicate who the final decision-makers were, which created some confusion and tension for team members. Finally, we focused on the process of the CER while giving less attention to how our report and program may ultimately impact the environment, people's lives, and the energy mix. These challenges are described below with a discussion of how principles from collaborative governance (CG) can help to address and improve these common challenges associated with community-engaged research. While these challenges are not novel among CER projects, we show how they play out in a new context (community solar program development), and we believe that drawing on principles from collaborative governance can help to address them for various CER projects ranging from socio-ecological to socio-technical systems.

*5.1. Identifying Community Participants*

A key issue in community engaged research is dealing with the complexities of defining communities and subsequently identifying which members to include in the research and decision-making process. Collaborative governance reconciles this complication by suggesting that inclusion should be based upon impact, such that all parties affected by a decision should have a say in decision-making. This usually takes the form of a representative to speak for themselves, an agency, a business, a community, or a large group of public stakeholders (Emerson et al. 2009). The key piece is to ensure that not only do the impacted parties have a voice, but the relative power of collaborators does not outweigh or become railroaded by the other in agreements or collective decisions.

Drawing on CG's principles for defining community partners, CER might first consider who is impacted and then strive to incorporate representatives of all impacted groups into the partnership team. This is important for propelling the collaboration towards a more democratic process. Leaving out impacted community members can affect the legitimacy (Johnston et al. 2010) of the project, ultimately influencing its viability. CER is good at collaborating with well-organized community groups, but often falls short of incorporating others who may be most vulnerable (Tumiel-Berhalter et al. 2005). In CG, this vulnerability is viewed as a disparity of resources, such as funding, time, expertise, and even power (Bryson et al. 2006; Huxham and Vangen 2005). CG utilizes facilitated leadership to (a) prevent any one party from exercising power over the others (Chrislip and Larson 1994; Bryson and Crosby 1993; Huxham et al. 2000) and (b) push the collaboration to redistribute and share the resources for the common vision or goal of the group (Emerson et al. 2011; Milward and Provan 2000).

Previous studies grapple with defining the community by shared geography, demographics, or sense of identity (Agrawal and Gibson 2001; Long et al. 2016). Geographic communities can be defined by physical boundaries, such as streets or landmarks (Burns et al. 2011). In our community solar study, we defined the community based on electric utility area geographic boundaries, which were superficially inclusive of all impacted parties (utility customers of the utility considering the community solar project). Still, the core research team only included village managers (who also control the municipal utility), a regional planner, academic scientists, and a representative of the energy provider. The general public (customers) were not key participants in the full research or implementation process, but rather were invited to participate at key stages in the research (answering questions, sharing attitudes/feelings, and interpreting results). Even then, the public's voice was heard and considered, but they were not well-represented in decision-making, other than by the village managers, who also had other concerns (managing the utility). In this sense, our community-engaged research fell short in its inclusiveness of community members. Our community participant definition was skewed towards authority figures that may make decisions based on a different set goals, values, priorities, and perspectives (Israel et al. 2005) compared to all impacted community members. Following CG principles would have offered the team a clearer framework and reasoning for including additional community voices as members of the core CER team and better allowed all impacted community

members a voice, thus improving the democratic nature of the decision-making process and bolstering the legitimacy of our research findings.

*5.2. Sustaining Participation and Avoiding Exploitation*

Some research projects have difficulties sustaining inclusive engagement. This is especially true in disadvantaged communities with a history of exploitation, including tribal communities (Morris 2017; Bullard 2008). Tribal communities have a long history of exploitation by state-based and academic authority figures, which can complicate establishing collaborative research projects (Doherty 2007; Smith 2013; LaVeaux and Christopher 2009). CER aims to avoid exploitation, and is specifically designed to incorporate community members into the research process rather than taking the knowledge gained elsewhere. Still, CER projects may inadvertently exploit community members by requiring considerable time and energy, often without corresponding compensation.

Drawing lessons from collaborative governance, CER projects should explicitly incorporate compensation for community members into project design and budgeting. Some CER projects already do this (i.e., Black et al. 2013); however, it is not well-established in CER practice. Incentives, such as proper compensation for time invested, must exist for impacted members to participate and collaborate in the process. A CG approach would suggest creating a fully compensated position to facilitate dialogue between the represented group and collaborators, as well as provide expertise and guidance to the represented group (Page 2010; Bingham 2009).

Following CG strategies with regards to power dynamics might also prove useful for helping to sustain community participation in CER projects and avoid exploitation. CG recommends understanding the system context, such as power dynamics (Ansell and Gash 2008) and/or historical trust and conflict issues (Radin and Romzek 1996; Thomson and Perry 2006). Understanding the system context can help to see what dynamics might emerge and further initiate the direction of the collaboration towards decision-making based on mutual trust. Applying this to CER, the community trusts the representatives to make decisions reflecting their needs, values, and goals. The partnership remains, and the community and decision-makers continuously work together to improve those decisions.

As mentioned above, some CER projects have used compensation to sustain participation (Israel et al. 2005). Others emphasize empowering community members to develop control over the research process as a way to sustain participation (Israel et al. 2001). Members of our core project team participated, at least in part, because it was part of their job. Team activities fit well enough with formal work responsibilities that they could participate as part of their regular work day. For this reason, sustained participation and commitment were not particularly problematic. Still, village managers did, at times, struggle to find time to devote. Village managers faced multiple competing demands, which presented a challenge to participation; however, the team maintained expectations that participation was part of the community partner's role.

Soliciting broader community participation for community meetings, however, proved more difficult. The Villages of L'Anse and Baraga had differing levels of community participation regarding the community solar program design. More L'Anse community members attended the community meetings, while the Baraga survey had a larger response rate. One reason for this could be our discovery of a recent example of exploitation in these communities. Our community research was conducted while, simultaneously, community leaders and the broader community were at odds regarding a large wind development project. The community leaders supported a wind development project in the area, yet the broader community opposed the project as they expressed skepticism of the wind developer's motives and feared being exploited by the large, external development companies. Our team felt that building trust, creating an open dialogue, and otherwise providing opportunities to empower locals to speak up about the project would be enough to bring community members to table. While we did not provide adequate compensation for all the time and effort that was necessary to participate in our research project, we did incentivize participation at community meetings by

bringing food, door prizes, such as LED light bulbs, and a raffle for a larger energy efficiency appliance. Due to the project's timeline, each meeting and survey occurred at different times of the year, which contributed to community members' conflicting schedules and prior commitments to community events. Our participation levels might have increased had we involved more general community members as representatives. These representatives could have been compensated for their time on this project. Additionally, the team could lean upon each representative's social network to recruit community meeting participants.

*5.3. Turning Research into Action: Power to Make Decisions*

While action towards social change is a goal of the research process (Stoecker 2012; Bradbury-Huang 2010), CER principles themselves provide little direction on how to take this research and use it to make decisions. We experienced several facets of this (discussed below) with our community solar project. Collaborative governance attempts to make the decision-making process more inclusive and more localized. We can use the CG literature to remind community-engaged scholarship that there are some people with final decision-making authority (Newig and Fritsch 2009), which requires transparency for the community to understand who that will be. Specifically, the involvement and power of these actors in decision-making may outweigh community suggestions and community energy desires.

We utilized the CER methodology in both villages to provide the opportunity for all village utility customers to give feedback about the community solar program's design. We planned to use this information to determine if the community should move forward with the program. For example, with the L'Anse community solar case study, the CER results suggested that the community wanted to move forward with a community solar project. Community members were interested in creating a community advisory board to oversee the program. Survey results indicated that multiple financing options (including a no down payment option) would function to increase community solar access to all community members, including low-to-moderate income populations. Suggestions also included a donation model either from community members outside the utility service territory or more affluent to less affluent community members. This methodology appeared to place power in community members designing the program for their community. Still, the ultimate decision-making process proved to be less democratic and transparent. Community members did not make a collective decision to move forward with the community solar program; rather, the village utilities and the energy provider had the final say. The factors, timing, and process of the post CER decision-making was not communicated to team members or the community. While the research team attempted to organize member participation across research, government, and private sectors, we mainly partnered with people in official power roles rather than the broader community. Because of this, we ultimately failed to explicitly outline decision-making as a series of iterative points and to clarify exactly who held decision-making power at each of these points.

Collaborative governance suggests that an inclusive, participatory strategy involving multi-level actors is effective for decision-making. A key component of this process is transparency (Ansell and Gash 2008). This includes communicating transparent ground rules and the shared vision with team members and all impacted community members to have the same expectations about final decision-making. Some CER findings (Israel et al. 2005; Johnson and Johnson 2003; Schulz et al. 2002) deal with transparency by including communication strategies external to community meetings (in the form of email, telephone, mailings, minutes, etc.). Utilizing CG strategies would have led us to do things differently in L'Anse and Baraga. Ideally, we would have included all affected community members in the final decision-making process. A step down from this involves transparency. While it is extremely difficult to provide information about every detail in the decision-making process, we should have been forthcoming about how and why these decisions were made. Being explicit about the incorporation of CG principles into the design of this community-engaged research project would have likely improved processes and outcomes.

*5.4. Focus on Outcomes Versus Outputs*

Community-engaged research places great emphasis on the process portion of conducting research, with the ultimate goal of facilitating social change (an outcome). However, integrating specific outputs and outcomes (evaluating the change) into the research design is not a well-established practice in CER. CER projects often emphasize community empowerment as an outcome (whether or not the team is successful in improving conditions), but neither are commonly assessed. Focusing more explicitly on outputs and outcomes may help teams to achieve this social change. CG projects are driven by an intended output, and communicating about the intended outputs (deliverables) is critical from the planning phase. Thomas and Koontz (2011) indicate that many projects that lack agreement on a shared vision from the project's beginning result in unfocused or incomplete outputs. The CG literature speaks to finding a shared motivation as key in communicating the goals (outputs). Committing to the process, outputs, and outcomes can help keep the researchers accountable.

With the L'Anse and Baraga project, our deliverable was a report for the utility and village council to make a final decision regarding the community solar program. Team members focused on the process of ensuring that the project would keep benefits local and improve quality of life by empowering community members to participate in the project as well as producing the final report. We did not build an assessment of how well our research process achieved these aims following our report and the subsequent program implementation. We could have utilized an assessment based on factors such as improved knowledge or clarity on key issues, perceived legitimacy of the project, improved trust, and how deliberations and final decisions were perceived (Emerson et al. 2011). In the planning phase, the team could have discussed and agreed upon the best way to operationalize and measure these outcomes. Including some sort of evaluative measure of outcomes of our work could help to demonstrate the real impacts of our research and open opportunities for adaptation to improve the process and program in favor of community needs.

## 6. Conclusions

In this paper, we reflect on our experiences using community-engaged research practices in one particular case study examining the potential of community solar projects in two communities. We also recognize that the challenges we experienced may not be novel, but that our reflections here build others' findings while being unique due to the community solar context. This conceptual reflection indicates four possible ways of improving community-engaged research by borrowing from research on collaborative governance. First, while community-engaged research projects often grapple with the complexities of defining communities and identifying appropriate members of communities to include in collaborative research, collaborative governance suggests that inclusivity should be broadly based on impact, in that anyone who will be impacted by a decision should have a seat at the table for decision-making. This can help to circumvent some of the conceptual challenges involved in defining community while also providing a question for ground truthing community engagement to ensure that the partnerships are inclusive of everyone who will be impacted by a project.

Second, sustained inclusion and engagement can be a challenge for community-engaged research; collaborative governance suggests the imperative of structuring participation to ensure that participants are compensated for their involvement and that their voices are empowered as equitable decision-makers on the team. For collaborative governance, this may mean involving those who are paid to be stakeholder representatives or may mean only involving those who have a professional or personal stake in the decision; for community-engaged research, this may mean balancing the possibilities of stakeholder representative involvement or providing compensation when asking those who have no professional duty to be involved.

Third, community-engaged research is often focused on ensuring just and inclusive engagement with the research process, while tending to ignore ultimate decision-making and the power differentials that shape it. Collaborative governance principles are attentive to ensuring that final decision-making

power is established and communicated clearly prior to deliberative activities, a lesson that community-engaged researchers may benefit from bearing in mind.

A fourth, final, and related point is that community-engaged researchers' focus on a just and inclusive process may be overshadowing the need to also have inclusive conversations about the intended outputs and outcomes of community-engaged research; collaborative governance research may also struggle with operationalizing and measuring outcomes but is more attentive to establishing shared understandings of intended outputs. Processes, outputs, and outcomes are all important, and attention to each may be improved by also paying attention to—as indicated by the lessons offered by collaborative governance—decision-making power, forms of compensation that can be offered for involvement and the diverse forms of representation that involvement can take, and the necessity of being both inclusive but also pragmatic in defining communities for research.

**Author Contributions:** Conceptualization was conducted by E.P., R.W. and C.S. Methodology was discussed and designed by E.P., R.W. and C.S.; Formal Analysis was conducted by both E.P. and R.W. Investigation and data collection was mainly conducted by E.P. with assistance from R.W. and C.S. Resources were collected by E.P. with contribution from R.W. E.P. prepared the original draft. Writing-Review & Editing was conducted by E.P., R.W. and C.S.; E.P. generated the visualization from Google Maps.

**Funding:** This research was partially supported by the Michigan Department of Agriculture and Rural Development, grant #791N7700467 and the American Public Power Association, grant #CG-2135.

**Acknowledgments:** We thank our team members, Brad Barnett, Brett Niemi, Jay Meldrum, Robert LaFave, and LeAnn LeClaire, who provided insight and expertise that greatly assisted this research. We would also like to show our gratitude to the Village of L'Anse and Village of Baraga community members for sharing their knowledge, insights, and concerns during the course of this research.

**Conflicts of Interest:** The author declares no conflict of interest.

## Appendix A

Appendix A is a compilation and analysis of notes taken during the Village of L'Anse community meeting.

**L'Anse Community Solar Feasibility Study**

**Project Report: Community Meeting**

**Introduction**

This report summarizes the process and results of a set of focus group discussions that were held at a community meeting at the L'Anse Area Schools building on 22 August 2017. The meeting was hosted by the Upper Peninsula Solar Technical Assistance and Research Team (UPSTART), which includes the Village of L'Anse, the Western Upper Peninsula Planning & Development Region (WUPPDR), WPPI Energy, and Michigan Technological University. UPSTART is evaluating the social and economic feasibility of implementing a community solar project in L'Anse.

The purposes of this meeting and associated focus group discussion were to share some basic information about community solar and a proposed community solar project in the Village of L'Anse with the broader community and to gain insight and understanding into how L'Anse area community members feel about the possibility of beginning a community solar project in their village and to uncover issues and considerations that could impact project adoption in L'Anse. We wanted to give community members an opportunity to learn about the project idea, to discuss it among themselves, and to share feedback about the idea with the project team.

This report includes a brief summary of methods and results. It concludes with a discussion of implications of these findings for the project team's continuing work.

The findings presented here are not based on a representative sample of L'Anse community members. It is highly likely that residents and businesses owners who are already interested in solar/community solar would be more likely to attend a meeting such as this. Also, we know that some of the participants who attended the meeting are not L'Anse utility customers (reside outside

Village limits) but are people who are particularly interested in energy issues. For these reasons, the ideas presented here should be considered as important ideas and themes that will likely continue to come up, but not as representative of the general attitudes of the community as a whole.

**Methods**

The community meeting was held at the L'Anse Area Schools Cafetorium on Tuesday, August 22nd from 6:30 p.m. to 8:00 p.m. The authors facilitated the meeting. Forty-nine participants attended the event. The participants sat in small groups of 5–6 people each at round and square tables spread around the large room. The facilitators began the meeting with a 15 min presentation about community solar and the proposed project in L'Anse. They then answered questions raised by the community.

Focus groups discussions lasted for 60 min following the initial presentation. The facilitators posed the following questions to the small groups and asked them to discuss each for about 5 min within their table. Also seated at each table was a member of the research team whose primary purpose was to take notes on the discussion. These notes were collected by the team and form the basis of data that were later coded for key themes. After discussing all five questions, the facilitators asked participants to publicly summarize key points they had discussed and wrote these important points on flip charts at the front of the room. These key points that were recorded form a second set of data collected at the meeting for analysis.

The five questions posed to participants were:

1-   What do you like about the idea of L'Anse doing a community solar project?
2-   What concerns you most about this idea or makes you think it might not work?
3-   Would you purchase shares? And why or why not?
4-   What are some things that the team needs to consider in designing the program?
5-   Do you think that L'Anse should move forward with this? Why or why not?

The author read through all of the notes that were collected at the August 22 meeting and coded them for key themes that emerged.

**Results**

Overall, participants were positive about the Village of L'Anse moving forward with a community solar project. Most participants were excited about the potential project and saw it as a positive development for the community, but some were also skeptical about some aspects of the idea. Participants raised several important considerations that could impact program design and project adoption in the local community.

Major themes that participants brought up are highlighted (in bold) and described in some context in the summary that follows. These themes are then discussed in the summary section.

**Question 1: What do you like about the idea of L'Anse doing a community solar project?**

L'Anse community members liked that the potential community solar project could help create a more sustainable energy program. It would allow L'Anse to be on the cutting edge of decreasing their **environmental** footprint. L'Anse can pave the way to greener energy sources through this type of **forward thinking**. By owning a more sustainable electric power source, L'Anse will have **local** accountability; i.e., by having local maintenance but also by steering away from unhealthy biomass burning. Benefits will be **economic** in nature. This type of investment will potentially add value to property. The benefits will also stay local, will be more **inclusive** of low-to-moderate income individuals, and ultimately will allow the community to come together and increase local pride. Additionally, if this project is a success, community members would like to see this project inspire and be replicated in other small towns. Additionally, due to the nature of community solar (optional purchase of shares) community members maintain **freedom to choose**.

**Question 2: What concerns you most about this idea or makes you think it might not work?**

Participants raised several concerns primarily with the soft design components. They were interested in several specific aspects of how the program would be designed and carried out. These included:

- Transfer of shares: Can they be bought and sold and under what conditions? If you were moving out of Village limits, could you easily get rid of your shares and be compensated fairly? What if someone who owns shares passes away? Could they "will" the shares to descendants or non-profits?
- Dibs on Purchasing: Who is allowed to purchase shares, when, and how many can be purchased. What if a few organizations or individuals buy up all the shares before others have a chance?
- Shareholder Liability: Participants were concerned with any liability that would come with purchasing a share in the community solar program. Community members felt that if by purchasing a share they might become responsible for any negative impacts or harm caused by this project. Will the Village take on this responsibility?
- Financing: Participants expressed interest in multiple finance models. Some felt that it will be really important to offer multiple options for payment and financing, as some may want to pay up front while others will need an easy financing option. It would be nice if the financing option could help people to improve their credit (count for their credit score). Would there be some kind of a bonus or reward for those who purchase shares early?
- Payback Period: The payback period was considered long to pay off the initial investment. There was also some question about the length of the term of service (20 years, 25 years, etc.): why would the credits stop at this point?
- Overall total participation: First, community members were concerned there would not be enough overall interest in the program, making it a waste of time. Alternatively, linked to the above "dibs on purchase" concern, community members felt the program would be sold out, before allowing all community members a chance to buy into the program. Will there be a possibility for expansion to accommodate all interest?

Participants did raise some concerns about technical design components, but these were less important to the participants. They generally accepted that community solar is technically feasible in the L'Anse area. The "technical" concerns that were raised included several things that might be considered part of the soft program design as well. These include the following:

- Size of the System: The community members were concerned with how the system's size would be determined. Linked to this, some felt the system might be too small, while still others felt the system would be too big. These concerns were based on the perceived interest in the program. Will the size be determined based on energy needs of the community? Estimation of participants following the feasibility study? And will the system be tailored to that interest?
- Maintenance: Who would be responsible for maintaining the system? Could local workers be used? Who would pay for maintenance?
- Vandalism: Participants noted that a similarly sized solar system in a neighboring community had experienced some vandalism. What would be done to reduce vandalism? And who would take care of it if it occurred?
- Selling the Energy: participants raised the question of whether we could guarantee that WPPI Energy would purchase the power generated, how this relates to the size of the system, and whether the Village would have control to sell the power generated to any other entity if they should so choose.

**Question 3: If this happens, do you think you will buy one or more shares for your home/business? Why or why not?**

The main answer was yes. Some groups felt they would buy into this program as it increases the **community empowerment and pride.** When asked why or why not, focus groups tended to

provide answers with "depending on . . . ". They steered back to concerns they had with designing the program. **Transferability** was still the overarching concern found with this question. A couple of incentives were offered as possibility to program design: if program managers offered a discount to community members who purchased shares by a certain date; a hard commitment from WPPI to pay back for power generated. This program would appeal to community members, especially **low–moderate income** if it were in place to help them build their credit back.

Those that answered no look at the program versus other "better" investments/returns on investment. For example, the returns with this program are not as high as investing in **energy efficiency**, such as insulation. Alternatively, some seemed to **lack trust** in the local government and local energy firms. These groups were concerned with the hidden benefits both the village and WPPI Energy would receive from this program.

**Question 4: What are some things that the team really needs to consider in designing the program?**

The following is a list of considerations that the UPSTART partners should include when designing a program for this community. Again, the considerations were focused more on the soft design components, including financing, transferability, and local/community benefits:

- Transferability: defining this clearly
- What to do if the shares sell out?
- Selling shares to non-profit entities
- Default on payback
- Defining how much a credit will be versus how much it is worth: dollar value that panels produce and then sell back to the grid
- Utilizing local workforce for maintenance
- Making it affordable for the community
- Multiple financing/payment options: depending on your economic/income status
- Trackable energy usage
- Utilizing some sort of trade system: i.e., work for share

Additionally, there is a question of different **consumer classes**. Some groups wondered how residential and business consumers would perceive benefits differently. The team should quantify the **true value** of environmental costs and benefits.

**Question 5: Do you think that L'Anse should move forward with this? Why or why not?**

Ultimately, "yes". People generally felt that they need more information regarding program design prior to committing to participating, but they were interested in seeing the project move forward and were glad to be involved in the discussion and opportunity. Community is more likely to be engaged if the ownership is kept **local**. This type of project helps the village look to the future energy needs and other environmental concerns. The village can help transition to other **alternative energy strategies** with this initial project.

**Summary and Implications:**

Respondents generally were positive about the idea of moving forward with the community solar project. They liked the idea for a combination of reasons primarily combining environmental benefits with economic returns and local empowerment. It is the wedding of these three major themes that seems most important as some participants will be more or less interested in each, but the possibility of bringing them all together seems like a real win.

The focus group discussions uncovered several important themes that the UPSTART team should consider in designing and marketing a potential community solar program.

**Local Control and Community Empowerment**

Community members really liked the idea of keeping the community solar ownership at a local level. This allows them to **think locally, but acting globally** to reduce their overall carbon footprint. The team should consider keeping as much of the process local as possible to keep this positive thinking surrounding the project. Local can help in multiple ways from producing the power locally, to having local accountability and a locally specific program design, to keeping the benefits local, to serving as a source of local pride and an opportunity for local education. Keeping the work local is important to the community. All of this increases community empowerment and pride in the community. The idea that L'Anse could be a leader for other communities and on the "cutting edge" is important. It could be seen as a leading UP community and a leading small community nationwide.

Another aspect of local control is having the freedom to choose whether to participate or not and to talk to local people who are designing the program. It is important for people to have this be a volunteer option that they are involved in the choosing of.

### Economics and Financing

The possibility of making an economic return on investment is important. While some were concerned about the payback time, most were attracted to the idea of making money from something like this. The UPSTART team needs to consider designing different financing options that are attractive to multiple economic class residents in the Village. Offering good financing models will be critical to the project's success.

### Fairness and Inclusivity

Community members felt it was extremely important that we continue to strive to find a program that especially includes low-to-moderate income individuals. This is directly related to the need for financing above. The team also might think creatively about other ways to involve lower income residents, including options for increasing credit rating or work-for-shares programs.

Participants also raised fairness concerns about who gets to buy shares, how many, and when. There was concern that a few individuals or businesses might buy out all the shares before others have an opportunity. The team should consider a sales model that provides an initial opportunity to the full community before opening up to purchase larger numbers of shares.

### Environmental Sustainability

Because solar PV technology produces fewer emissions, residents were excited about the prospect of producing energy from a source that is **environmentally sound.** This was even more prevalent when community members compared the potential project to the L'Anse biomass plant. Highlighting the environmental benefits of solar PV will help to make the project more attractive to residents.

By investing in a solar PV system, L'Anse residents can begin to help reduce the village's environmental footprint. Many community members liked this way of **forward thinking**. By doing something like this, community members could feel good about participating in a program that works to improve future generations' environment.

Some also expressed interest in alternative options for investing in reducing footprints and alternative source of energy. For instance, **energy efficiency** is also important and both reduces environmental impact and provides faster economic returns. The team should consider combining community solar with more comprehensive strategies for reducing energy consumption and/or other alternative energy options beyond community solar.

### Trust

The UPSTART team should provide transparency about benefits to residents and local government, as there appeared to be a lack of **trust** regarding the motives for completing a project like this.

### Guarantees

Participants want some guarantees on their return. They want details on transferability to knowing they can sell. The team should strive to make this as easy as possible and to facilitate it. They also wonder if there is a guaranteed return on investment and a guarantee from WPPI to purchase the power. This is also related to responsibility. Who will take care of things like liability, maintenance, and vandalism?

The team should consider clarifying these points and communicating them in marketing language.

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
