# Peer review of "Putting Research to Action: Integrating Collaborative Governance and Community-Engaged Research for Community Solar"

_socsci, doi:10.3390/socsci8010011_

Round 1

Reviewer 1 Report

There is increasing focus on conducting community engaged research beyond medical and public health fields. This reflective essay is timely as decentralized renewable energy strategies are gaining significance in both developed and developing countries and can benefit from community engaged research. The findings and contribution from the action research project using collaborative governance are clear and well stated. The contributions are useful for practitioners and policy makers and those who intend to perform community engaged research.

I have two comments:

 The findings themselves are not novel as there is ample evidence from studies conducted in the fields of socio-ecological systems and one can question if the challenges highlighted by the study are unique for socio-technical systems such as community solar.

The choice of collaborative governance vs. other approaches such as participatory design or collective impact is unclear. An upfront clarification of the other approaches considered but disregarded for these projects will make the case for collaborative governance stronger.

Regardless, community solar is expanding at rapid pace and methodological contributions highlighted by the essay are useful to facilitate social innovation.

Author Response

Dear reviewer 1,

Thank you very much for reviewing our manuscript. We would also like to thank you for your comments and suggestions on this paper. We have taken the suggestions and revised the manuscript accordingly. I am attaching a table that shows how we specifically addressed each suggestion.

We acknowledged in our results and conclusion section that the results may not be novel in a broader sense, but they are unique in the context of community solar. We added statements that acknowledge how others have dealt with these challenges and how we specifically dealt with the challenges.

Please find the attached table for specifics on each suggestion.

Thank you

Reviewer 2 Report

Reviewer Comments

“Putting research to action: Integrating collaborative governance and community engaged research for community solar”

This manuscript investigates community solar energy systems in Michigan, USA through a collaborative governance and community engaged research lens.  Community solar is an important and emerging renewable energy model, and the application of these theoretical frameworks is unique in this realm.  Using a variety of methods, the analysis finds that there are challenges in the decision making processes for community solar programs, such as administration, education, participation, communication, program design, and others, yet these frameworks may illuminate best practices for future programs.  The focus on community solar through these theoretical lenses is an interesting and understudied topic, so given this value-added and interest to a broad readership, I would recommend only minor revisions to make it publishable in Social Sciences.

Overall, this manuscript offers some intriguing analysis, but the writing could be streamlined and better crafted in some areas, especially in the background and literature section.  In particular, I recommend revisions in line with the following comments:

1.      The abstract and introduction sections are rather well done, but I think there needs to be a much deeper explanation of community solar as you transition into section 2.1.  As it currently stands, much of the narrative offers broad strokes on community energy themes, which is good, but the author(s) should take it one step further to explain the different types of community solar (i.e., utility-based, nonprofit, special purpose entity, etc.) and what is allowable in Michigan through state policy.  There is discussion about how community solar is often political and needs to understand local contexts, but the lack of enabling policy discussion here is an oversight.  Many states do not have formal laws on the books to enable community/virtual net metering for community solar, so the authors ought to discuss this and how it works in Michigan (such as if only innovative program designs have developed). There are also a few papers here you should cite, such as Klein & Coffey (2016) and Chan et al. (2017).

2.      The community engaged research and collaborative governance write-ups seem much too long for a background narrative / literature review section.  I would suggest cleaning this up a bit.  For instance, I do not see great value in having such long narratives (and posing numerous questions) around defining the community and explaining prior work on participation sustainability / exploitation.  Further, much of this is not specific to community/solar energy, so I feel like it would be better to streamline this and make it relevant to your study.  I know, as you’ve noted, that there is not a ton of literature on CER in energy matters.  However, when you have this narrative around accessing LMI populations, you should talk about DOE’s SunShot program and the Solar in Your Community Challenge, which explicitly targeted those populations.

3.      Similarly, the collaborative governance section is easy to get lost in and should be shortened a bit.  This lens is much more common in the energy realm, which is a positive attribute that you should discuss in greater depth.  From a general perspective, you should cite prominent works such as Innes & Booher (1999), Innes & Gruber (2005), and Margerum’s (2002) three-level model.  From an energy perspective, there are key works to consider as well, such as Purdy (2012), Pitt et al. (2017), and many others.  I don’t see much of a need for the final paragraph in this section.

4.      Good discussion of the communities, but this section would be enhanced with a visualization (map?) of the area to laypeople.  Also, you should note the particular insolation metrics (in kWh/m2/day) for this area of Michigan and the electricity prices (in cents/kWh).

5.      I think the mixed/multi method approach is interesting, and the authors do a good job of explaining these various research strategies, but it’s a bit unclear how they link together from a methodological perspective.  I wonder if these were enacted as a matter of convenience, as the links seem tenuous.  I’m assuming that the point here was to reach a larger number of folks from a variety of perspectives, but the authors need to do a better job at explaining why this multi-method approach was, in fact, the best research strategy.  Further, some of the methodological discussions lack specificity which hinders the replicability of the study.  How is LMI being defined here, for instance (HUD definition?)?

6.      As a general statement, too much of the discussion still talks about general themes and theories, and not enough about specific findings or implications for the Michigan research process.  The authors should work on streamlining the introductory narratives per each subsection and jump more quickly into the findings in their meetings and interviews.  As this is currently written, the reader is left with many questions about what the results were of the survey and interview process so eloquently described in the methods section.  I understand that this is a conceptual reflection, but I think it would be helpful, as an example, to better illustrate community solar program design options for these communities after interacting with the stakeholders?  What, in particular, were they saying (include quotes?)?  What worked/didn’t work for them (setting up LLC’s, NPO’s, crowdfunding programs, etc.?)?  This, to me, is the highest value add from this type of research paper, and I think the authors really need to work on making these findings more clear.  Data visualizations would add value.  Much of this comes forth in the appendices, but the main article text needs to include findings about payback periods, administration, project size, etc., if/when possible.

7.      The conclusion is quite long winded and should be split into a few paragraphs.  Again, the authors ought to focus more on the findings of their discussions with community members than the CER/CGT theories themselves, or at least better link the two. 

A few smaller points:

--There are many contractions in this manuscript, which I find inappropriate for academic writing.

--There are some statistics that need citation, such as at line 51.  Please check throughout.

--Be careful with the acronym use and using things like CER before they are defined (see line 62).

--Watch for comma use after introductory phrases, and around however terms (see line 87)

--Many of the in-text parenthetical citations are not listed in alphabetical order, which perhaps is a journal style guide preference, but it seems non-traditional to me.

--This wind project that was mentioned a few times ought to be described in greater detail.

--It might help to better explain the breakdown of electric utility types in Michigan (i.e., IOUs, co-ops, & munis).  Are there a lot of municipal utilities in this part of the state?  How do you see this playing into the local control and decision making process?

--I think the authors should explicitly outline a research question early along in the manuscript.

--Much of the writing could be streamlined.  Check for run-on sentences and superfluous terms.

Author Response

Dear reviewer 2,

Thank you very much for reviewing our manuscript. We would also like to thank you for your comments and suggestions on this paper. We have taken the suggestions and revised the manuscript accordingly. I am attaching a table that shows how we specifically addressed each suggestion.

We did want to address one specific suggestion in this cover letter. The suggestion was to include more empirical results from the actual social feasibility study we conducted. We intentionally left these type of results out of this paper. The purpose of the paper is to satisfy the call for the special issue of a reflection on community engaged research and how the process could be improved. Additionally, the actual feasibility study results are presented elsewhere so we did not include them here. We tried to address the suggestion by better clarifying in the paper that this was a reflection piece rather than a presentation of empirical findings.

Please find the attached table for specifics on each suggestion.

Thank you
